# Farm-Level Risk Factors of Increased Abortion and Mortality in Domestic Ruminants during the 2010 Rift Valley Fever Outbreak in Central South Africa

**DOI:** 10.3390/pathogens9110914

**Published:** 2020-11-04

**Authors:** Melinda K. Rostal, Sarah Cleaveland, Claudia Cordel, Lara van Staden, Louise Matthews, Assaf Anyamba, William B. Karesh, Janusz T. Paweska, Daniel T. Haydon, Noam Ross

**Affiliations:** 1EcoHealth Alliance, New York, NY 10018, USA; karesh@ecohealthalliance.org (W.B.K.); ross@ecohealthalliance.org (N.R.); 2Institute of Biodiversity, Animal Health and Comparative Medicine, College of Medical, Veterinary and Life Sciences, University of Glasgow, Glasgow G12 8QQ, UK; sarah.cleaveland@glasgow.ac.uk (S.C.); Louise.Matthews@glasgow.ac.uk (L.M.); Daniel.Haydon@glasgow.ac.uk (D.T.H.); 3ExecuVet PTY LTD., Bloemfontein 9301, Free State, South Africa; execuvet26@gmail.com (C.C.); laravst@gmail.com (L.v.S.); 4Universities Space Research Association, Columbia, MD 21046, USA; assaf.anyamba@nasa.gov; 5NASA Goddard Space Flight Center, Biospheric Sciences Laboratory, Greenbelt, MD 20771, USA; 6Centre for Emerging Zoonotic and Parasitic Diseases, National Institute for Communicable Diseases, National Health Laboratory Service, Johannesburg 2192, South Africa; januszp@nicd.ac.za

**Keywords:** Rift Valley fever, abortions, deaths, Bayesian, clinical signs, risk factors, prediction, farm level, ruminants, hurdle multinomial

## Abstract

(1) Background: Rift Valley fever (RVF) outbreaks in domestic ruminants have severe socio-economic impacts. Climate-based continental predictions providing early warnings to regions at risk for RVF outbreaks are not of a high enough resolution for ruminant owners to assess their individual risk. (2) Methods: We analyzed risk factors for RVF occurrence and severity at the farm level using the number of domestic ruminant deaths and abortions reported by farmers in central South Africa during the 2010 RVF outbreaks using a Bayesian multinomial hurdle framework. (3) Results: We found strong support that the proportion of days with precipitation, the number of water sources, and the proportion of goats in the herd were positively associated with increased severity of RVF (the numbers of deaths and abortions). We did not find an association between any risk factors and whether RVF was reported on farms. (4) Conclusions: At the farm level we identified risk factors of RVF severity; however, there was little support for risk factors of RVF occurrence. The identification of farm-level risk factors for Rift Valley fever virus (RVFV) occurrence would support and potentially improve current prediction methods and would provide animal owners with critical information needed in order to assess their herd’s risk of RVFV infection.

## 1. Introduction

Rift Valley fever virus (RVFV) is a zoonotic arbovirus that can cause widespread outbreaks interspersed with periods of very few clinical cases. Outbreaks of Rift Valley fever (RVF) have occurred across Africa and some parts of the Middle East, with eastern, southern and western Africa reporting repeated widespread outbreaks [1]. Herds affected by RVF can suffer high mortality rates in young lambs (~90%), abortion rates of up to 100% among pregnant domestic ruminants, and mortality in adult sheep (~20%), cattle and goats (~10%) [2,3]. During a large outbreak, a farmer may lose nearly an entire birth cohort for a season [4,5], resulting in significant economic losses. Though RVFV vaccines are available, many farmers cannot afford them or choose not to vaccinate believing their farms are not at risk [6] given the long periods between RVFV outbreaks (~2–36 years [1]).

The ecology of RVFV results in an ongoing cycle of endemic and epidemic periods, understanding this allows the identification of several potential risk factors for RVF outbreaks. RVFV is a mosquito-borne virus of ruminants [7]. During the endemic cycle, it is hypothesized that a low level of RVFV transmission causes seroconversions and small, undetected outbreaks of RVFV [8,9] between epidemics. The endemic cycle is maintained through transovarial transmission in certain floodwater mosquito species (e.g., *Aedes* spp.) whose breeding habitat is restricted to shallow depressions, or pans [7]. It is likely that livestock- and cryptic wildlife-vector transmission is sufficient to maintain the virus [10,11]. Although many species of wild ruminants appear to be susceptible to RVFV infection, the role they play in RVFV maintenance is unknown [12,13]. The epidemic cycle of RVF, causing large and widespread outbreaks, is thought to be driven by environmental conditions. This includes flooding caused by abnormally high and prolonged precipitation events [14,15] that support massive breeding of mosquitos that are competent horizontal vectors of RVFV (e.g., *Culex* spp.), which amplifies RVFV infection in susceptible animals.

Epidemiological investigations have linked RVFV emergence in new countries to domestic ruminant trade [16,17,18]. This was further supported by phylogenetic analyses of RVF outbreaks in endemic countries [19,20]. RVFV seroprevalence studies have also identified trade-associated risk factors [21,22]. The literature is dominated by serological studies, which can identify animal- and farm-level risk factors for RVFV exposure, but different factors may affect the risk of RVF outbreaks on a farm. This could reflect the distinctive patterns of RVFV infection that occur during endemic and epidemic cycles. At a larger scale, spatiotemporal analyses on the World Animal Health Organisation (OIE) RVF case data have identified ecological risk factors (e.g., precipitation levels [23], vegetation changes [14], landscape type, and wildlife presence [24]) and regions at higher risk for RVF outbreaks [23,25]. Other studies have modeled the spatiotemporal patterns of RVF outbreak spread among farms [26]. While some of these prediction systems have successfully predicted regional RVF outbreaks [27], they have not performed well in all regions [28], including in South Africa. These regional predictions do not have sufficient resolution for farmers to understand their risk of an RVF outbreak with respect to their neighbors’, due to differences in local conditions and farm-level risk factors. As individual farmers are the first line of control of RVFV, it is critical that their farm-level risks for an outbreak be evaluated.

Here, we investigated RVF occurrence, severity, and risk factors on farms in central South Africa, a region with a recent history of RVF outbreaks. We hypothesized that the occurrence of RVF on a farm in 2010 would be associated with risk factors purported to maintain the RVFV endemic cycle as well as with the ruminant trade. We further hypothesized the severity of a 2010 outbreak on a farm, as measured by the reported number of abortions and deaths, would be associated with risk factors that support vector amplification and host susceptibility. We administered a questionnaire to farmers to collect farm-level data on RVF impacts and risk factors. We developed a Bayesian multinomial hurdle model to assess the risk factors associated with farm-level RVFV occurrence and outbreak severity in 2010. We found strong support that the proportion of goats in the herd, the number of water sources, and the proportion of days with precipitation (January–March 2010) were positively associated with higher severity (numbers of deaths and abortions). We found that short-distance transmission between farms is important but more evidence is needed to identify specific risk factors that would allow farmers to assess their farm’s specific risk of an RVFV occurrence during a period of elevated risk.

## 2. Results

### 2.1. Farm Characteristics

One hundred twenty farmers completed questionnaires comprising questions about their farms, stock, and losses during 2010 when South Africa experienced widespread outbreaks. The median number of owned domestic ruminants reported by participating farms was 365 (range: 5–6500). Forty-four (37%) farmers reported that an RVF outbreak occurred on their farm in 2010, but most were not laboratory-confirmed (Table 1). On those farms affected, the median farm-level estimate of the percent of ruminants that had an abortion was 4% (range: 0–71%) and the median percent that died in 2010 was 3% (range: 0–60%) (Figure 1). Among farms that reported that sheep (adults and lambs) were affected, the median number of sheep on a farm that aborted during 2010 was 15 (range: 0–468) and the median number of sheep and lambs that died was 10 (range: 0–150).

### 2.2. Factors Associated with the Presence of Rift Valley Fever Virus on the Farm

We used a Bayesian multinomial hurdle model to estimate the effect that various risk factors (Table 2) had on the number of deaths and abortions that occurred on each farm (*n* = 120) during the regional RVF outbreak period in 2010. The model had two components that provided a joint estimate of the results. The first component used a binomial regression as a hurdle that estimated the probability of RVF occurring on the farm. The second component of the model used a multinomial logit that estimated the number of animals on the farm that had no clinical signs of RVF, had abortions, or died, conditional on RVF occurrence. The probability of RVF occurring on a farm was modelled as a latent variable and estimated by the hurdle portion of the model. All of the parameter estimates predicting RVF occurrence had very wide credible intervals (Figure 2c (standardized) and Table 2 (unstandardized)). The proportion of the posterior that was positive was estimated for each parameter: the cumulative precipitation between 15 October and 15 December, 2009 (0.38); whether the livestock mixed with wildlife (0.20); the farm size (0.26); the number of pans accessible to the ruminants (0.93); whether the farm was a semicommercial (0.84) or commercial (0.80) farm; the distance from which new ruminants were purchased (0.93), and number purchased (0.05; Figure 2c).

A Gaussian process (GP) was included in the linear predictor for whether RVF occurred on the farm to account for spatial variation between the farms. In our system, 99% of the covariance in the model occurred at distances <5% (13.7 km) of the maximum distance between farms (288 km; Appendix A). As the mean distance between farms was 115 km, only 1% of pairwise distances (*n* = 7140) among the 120 farms were less than 13.7 km.

The wide credible intervals provide poor support for the risk factors evaluated. Some of the trade predictors had low sample sizes, further data and analysis may allow better estimation in the future. There is stronger but still ambiguous evidence (93% of the interval > 0) for the influence of the number of accessible pans and the distance from which livestock are purchased. This ambiguity leaves farmers without strong evidence for farm-level risk factors by which to evaluate their own risk. The GP suggests that neighboring farms were at a higher risk of having RVFV if they were within 14 km of an affected farm.

### 2.3. Factors Associated with the Number of Deaths and Abortions on the Farm

The severity of an RVF outbreak as measured by the number of animals that had abortions, that died, or that had no clinical signs of RVF was estimated by the multinomial portion of the model. The parameter estimates of the linear predictors for both abortions and deaths had well-defined posterior distributions (Figure 2). The parameter estimates for the proportion of sheep, goats and days with precipitation (15 January–15 March 2010) as well as the number of water sources on the farm were positively associated with the outcome for both deaths and abortions. In contrast, the two-month cumulative precipitation and the total number of ruminants on the farm were negatively associated with both abortions and deaths. The effect of vaccinating before the outbreak was negatively associated with the number of deaths, but not the number of abortions. Although all effect sizes were small, the parameter estimates of RVF abortions had an effect size of a larger magnitude than the effect size of the same parameter on the number of deaths, except for vaccination (Figure 2). We also estimated the predicted probabilities of the multinomial outcomes (abortions and deaths (Figure 3) and no clinical signs (Appendix A)) for the continuous variables. We found that the predicted probability of death and abortion (Figure 3) aligned in direction and scale with the median parameter estimates of the number of deaths and abortions (Figure 2) for all variables, except for the effect of proportion sheep on the number of RVF deaths (Figure 3a). If sheep represented 0.1 or less of the total ruminants on the farm, the probability of a death during the RVF outbreak was approximately 0.06, whereas if only sheep were owned the probability of death was 0.03.

These results suggest that all of the variables included in the model are farm-level risk factors for the number of deaths and abortions during an RVF outbreak; however, the small effect sizes suggest that there may be additional risk factors.

### 2.4. Model Assessment

The estimates of all parameters in the Bayesian model converged (Appendix A), and the scale reduction factor was sufficient (1) for all parameters. We tested models with slightly different structures (Appendix A), but they were not significantly different based on the Watanabe-Akaike Information Criterion (WAIC) (Appendix A). The selected model explained 49% of deviance.

We evaluated the full and partial effects of the model and compared the model predictions of the number of deaths and abortions to the data reported by the farmers (Appendix A, respectively). No spatial clustering was visible within the residuals (Appendix A). Predictability increased if the hurdle linear predictor (with the GP) was included, though it remained around 50% (area under the curve (AUC) of 0.52).

## 3. Discussion

Here, we tested the hypotheses that severity, the number of deaths and abortions, of an RVF outbreak was driven by risk factors linked to host susceptibility and vector amplification and that the occurrence of RVFV presence on a farm was driven by risk factors linked to trade (importation) and to the RVF endemic cycle. We found that most farmers reported lower rates of deaths and abortions during a period of high risk for RVF in 2010 than expected from the literature. We found strong support that the proportions of the herd that were sheep and goats, the number of water sources, and the consistency of rainfall on a farm were associated with higher numbers of abortions and deaths. We did not find strong evidence for potential risk factors of RVF occurrence on farms; further data on these and other factors are needed to predict occurrence. Small-scale spatial processes are important in the pattern of RVF occurrence; it is correlated in nearby (<14 km) farms.

### 3.1. Impact of RVF Outbreaks on Domestic Ruminants

We provided a questionnaire-based assessment of the impact of RVF outbreaks on individual farms. The median proportions of animals affected by mortality and abortion reported by the farmers during an RVF outbreak (3% and 4%, respectively) were extremely low given the high rates of RVFV-associated mortality in young lambs and abortion in pregnant ruminants reported in the literature [3,29,30]. However, some farmers did report losing up to 60% of their ruminants with up to 71% of their domestic ruminants aborting, more in line with the historical records and the results of experimental infection.

Mdlulwa and Ngawne [31] reported an overall loss of 22% of sheep flocks affected by RVF in 2010 (including 47 surveys that overlapped with our study area). Unfortunately, they did not report farm-level losses, instead reporting the total number of animals lost during the 2010 RVF outbreak. Nabeth et al. [32] estimated that across all farms surveyed during the 1998 RVF outbreak in Mauritania the abortion rates were 47% in cattle and goats and 10% in sheep. They also estimated peri-natal mortality rates of 47% in goats and 5% in cattle. Neither study provided per farm rates as we did, which are important for understanding the farm-level impacts of RVF. Using data reported to the OIE, Métras et al. [33] estimated the mean all-species, on-farm morbidity in South Africa in 2010 to be 7–9%, while case fatality rates were 66–79%.

### 3.2. Factors Associated with RVF Occurrence on Ruminant Farms in 2010

The estimates of the parameters from the hurdle portion of the model all had wide credible intervals, providing negligible support for an association with the occurrence of RVF. The small sample size likely contributed to the wide credible intervals for the risk factors associated with importing RVF (the number of ruminants purchased and the distance from which they were purchased). Of the 120 farms surveyed, only six farms reported that they purchased ruminants in the past 12 months. Previous studies identified trade as being associated with outbreaks during RVF emergence in a new country [16,17,18], which suggests that a larger sample size may have identified an association (93% of the posterior distribution for the variable, the distance from which animals were purchased, was above zero).

The self-classified production type may have had large credible intervals due to the variation in RVF risk across different sectors of commercial livestock farming. For instance, both feedlots and dairies would be considered commercial, however, feedlots are unlikely to include pregnant animals and the animals would be old enough that mortality would be unlikely, whereas dairies usually have a significant portion of their herd that is pregnant at any given time. Previous work by Mdlulwa and Ngwane [31] found that commercial farmers were more likely to be affected by RVF than smallholders. They attributed this to the fact that the state vaccinated the ruminants on many of the smallholder farms, whereas the farmers themselves handled the vaccination of their animals on the semicommercial and commercial farms.

The Gaussian process was included in the model to account for spatial variation, and most of the covariance explained by the GP was amongst farms that were within 14 km of another farm. This implies that spatial correlation occurs at a very fine scale during an RVF outbreak period. Although 14 km is short in comparison with the longest pairwise distance in our dataset (288 km), it is in line with Métras et al.’s [34] results. Métras et al. [26] found that the risk of RVF spread was higher between farms that were only 5 km and 15 km apart during the South African outbreaks in 2010 and 2011, respectively.

Among the variables associated with RVFV endemicity, none had well-defined posterior distributions despite having known ecological associations with RVFV. While 93% of the posterior distribution for the variable for number of pans was above zero, the results of the analysis suggest that either one or more important variables were excluded, the data were insufficient or that some (or all) of these variables do not have strong associations with the presence of RVF at farm level. If the case of the latter, then it suggests that farmers have very little information on which to assess the specific risk for RVF for their farm. Global analyses can and have been used to identify geographic regions that are at risk [27,35], but they lack sensitivity to local conditions and farm-level risk factors, especially in southern Africa [28]. In South Africa, following a warning about the high risk of RVF from the government and an isolated outbreak within the study area in April/May 2018 [4,36], 39% of farmers surveyed in August–December 2018 said they had not vaccinated in the past year [6]. The primary reason they gave for not vaccinating was that they did not perceive a risk to their own farm [6]. A significant amount of research has investigated risk factors of large-scale (regional) and microscale (individual animal) risk of RVF; however, few if any analyses have evaluated farm-level risk factors associated with an outbreak of RVF on a specific farm. We did not find strong support for the farm-level risk factors based on survey data alone. Further work to predict RVF outbreaks occurring on farms during periods of heightened regional risk is critical and would provide the basis for livestock owners to better manage the impacts of RVF.

### 3.3. Factors Associated with Deaths and Abortions among Domestic Ruminants during the RVF Outbreaks of 2010

We found strong support for an association between the risk factors and domestic ruminant deaths and abortions in 2010. Among the variables included to represent vector amplification, increased number of water sources and proportion of days with precipitation in early 2010 were both positively associated with deaths and abortions. This was consistent with previous work that linked RVFV seroprevalence with the presence of nearby water sources [22]. However, the two-month cumulative rainfall was negatively associated with both abortions and deaths, which was contrary to the positive association identified between high levels of precipitation for 60 days or greater and higher risk of regional RVF outbreaks [23]. This suggests that the relatively high cumulative precipitation across the entire region does not directly translate to risk at the individual farm level, for example, differences in soil types between farms may result in different flooding patterns and associated vector abundances.

Among the host susceptibility factors, the proportions of the herd that were sheep and goats were both positively associated with higher abortions. This is consistent with findings that adult sheep are more susceptible to RVFV infection than cattle or goats [3]. In contrast to the positive association seen for abortions, the proportion of the herd that were sheep was negatively associated with the number of deaths. This was surprising as young lambs are extremely susceptible to RVFV and adult sheep are more susceptible than other domestic ruminants [29]. This change may be related to the fact that among farms with a high proportion of sheep (>0.92), all but one also owned over 1000 sheep. Farms with large numbers of total ruminants were negatively associated with the number of deaths and abortions that occurred. This was also unexpected as large herd size had previously been associated with greater risk of herd seropositivity [37]. Among the 27 farms that owned >1000 domestic ruminants only five of those farms owned more cattle than sheep, which makes it less likely to be due to species differences in the severity of clinical signs. While these variables were not colinear, both may have been affected by a bias if workers on larger farms did not notice as many of the losses that occurred due to RVF while the ruminants were kept in the fields.

The use of an RVF vaccine prior to the outbreak was negatively associated with the probability of deaths during the outbreak. While this is an expected association, a study conducted shortly after the 2010 outbreak by Mdlulwa and Ngwane [31] did not identify an association between the number of sheep lost during the 2010 RVF outbreak and whether the farmer vaccinated their sheep against RVFV. Additionally, there was no association between whether a farmer vaccinated before the outbreak and the number of abortions. The Smithburn vaccine can cause abortion in pregnant ruminants. While we do not have data on the type of vaccine product used, if farmers vaccinated with Smithburn at the start of or during the outbreak on their farm (seven farmers stated they vaccinated both before and during the outbreak), they may have inadvertently induced abortions in their animals.

### 3.4. Limitations of the Study

Our questionnaires were conducted 5–9 years after the 2010 RVF outbreaks in South Africa, and thus could be affected by recall bias. However, given the community trauma that RVF caused during the 2010 RVF outbreaks, with over 14,000 cases in livestock and several human deaths [23,38], we believe that the farmers would still remember the deaths and abortions that occurred on their farm during that outbreak. However, the social trauma of the outbreak may have been significant enough that the farmers attributed any deaths or abortions that occurred during the period at high risk for RVF to have been due to RVF. Unfortunately, confirmation of the etiology of the outbreaks was impossible at the time of the questionnaires. This may explain why so many farms had a small percentage of affected animals. Alternatively, our estimates of abortion rates may have been skewed as we calculated them using the total farm-level population for that species, including male and young animals or by farmers that attributed RVFV as the cause when RVFV was not circulating on their farm. Finally, we did not obtain data on what type of vaccine the farmer used prior to the outbreak (e.g., Smithburn, Clone 13, inactivated, etc.), the Smithburn vaccine is known to be abortogenic (as discussed above), whereas the inactivated vaccine does not cause clinical signs though it requires a booster.

Without widespread surveillance before and during an outbreak we cannot discern whether some farms have ruminant herds that are much less affected than the literature suggests [30,39] and identify the most important farm-level risk factors for RVF occurrence. We recommend that future research, surveillance, and response efforts collect information that is difficult to obtain following outbreaks. Ideally, farms enrolled as sentinel sites prior to outbreaks would enable rapid assessment of previous exposure at the first notice of an RVF outbreak in the region (e.g., a small serosurvey). During an outbreak, efforts should prioritize collecting blood and tissue samples for molecular diagnosis on the enrolled farms that suspect an RVF outbreak and collecting climatic data at the farm sites (e.g., remote sensing). Immediately following the outbreak, the serosurvey could be repeated to estimate extent and the sentinel farmers could be administered a questionnaire to determine whether they vaccinated (brand, batch, lot, expiration date; species, age and sex of the vaccinated animals and when the farmer vaccinated them), record the number of deaths and abortions per farm (correlated with the diagnostic data), describe ruminant demographics, water sources, current biosecurity measures, and the mitigation measures used during the outbreak.

## 4. Materials and Methods

### 4.1. Study Area

This study was conducted in central South Africa within a 200 × 200 km area approximately between Bloemfontein on the east, Mokala National Park on the west, and 100 km north and south of those locations (Figure 4). The WGS84 decimal degree geographic coordinates of the study area corners were −28.20, 24.20; −30.20, 24.20; −28.20, 26.40; −30.20, 26.40 (Figure 4). The study area was selected as it included the epicenter of widespread RVF outbreaks in 1950–1, 1974–5, and 2009–10 [40]. The region is temperate and fairly flat with many pans that temporarily fill with water following precipitation [41]. The southern part of the study area is primarily used for sheep farming, and the northern portion has more crop (east) or cattle farming (west) [42]. The study area has been more thoroughly described previously [43,44].

### 4.2. Ethical Approvals

This study received ethical approval to conduct human subjects research using questionnaires approved by HummingbirdIRB (2014-25), the University of Witwatersrand (M140306), Northern Cape Provincial Health Research Ethics Committee (NC2015/001), Free State Provincial Department of Health (04/04/2015), and the U.S. Army Medical Research and Materiel Command, Office of Research Protections, Human Research Protection Office (A-20745.1a and A-20745.1b). A follow-up study was also approved and conducted via phone to ask a few additional clarification questions.

### 4.3. Survey Data Collection

The collection of the survey data has been previously described [43,44]. Briefly, geographic points were randomly selected using ARC-GIS 10.2 (ESRI, Redlands, CA, USA). The nearest ruminant-owning farm/household to the geographic point was identified and invited to participate. Following the informed consent process, the farmer manager or owner was asked to complete a questionnaire regarding the history of RVF outbreaks on the farm, RVFV vaccination, animal demographics, trade patterns, and farm management and biosecurity practices. These data were collected in 2015 and 2017 using Open Data Kit Collect [45] and have already been analyzed for their intended purposes [43,44].

A brief follow-up questionnaire was used to obtain additional information specifically about the demographics of the ruminant populations on that farm in 2010, clarify the number of deaths and abortions in 2010 and whether the farmer vaccinated against RVFV prior to the 2010 outbreak. This follow-up survey was administered by phone in 2019 to all previously participating farmers (2015/2017) who could be reached and agreed to participate. While we were able to obtain estimates of the total population size by species just prior to the outbreak in 2010, we did not obtain data on the sex ratio or age structure of the animals owned in 2010. The 2010-specific data were collected via the phone questionnaire, and the remaining risk factors on the farm were assumed to be the same in 2010 as when they were collected in 2015 or 2017.

### 4.4. The Precipitation Data

Daily precipitation data were extracted from the African Rainfall Climatology dataset from the NOAA-CPC archives from 1 January 2009 through 31 December 2017. The rainfall estimates are derived from METEOSAT cold cloud duration data adjusted with in situ rainfall gauge readings to produce a gridded layer of precipitation across Africa [46]. The resolution was approximately 10 km^2^. The two-month cumulative rainfall and the percent of days with any precipitation were calculated for the period of 15 January 2010–15 March 2010, and the cumulative rainfall was also calculated for 15 October–15 December 2009. All variables were assessed for co-linearity.

### 4.5. The Model

We used a Bayesian multinomial hurdle model to estimate the effect various risk factors had on the number of deaths and abortions that occurred on the farm during the 2010 RVF outbreak period. The model had two components that provided a joint estimate of the results (Figure 5). One component of the model used a binomial regression as a hurdle that estimated the latent variable representing the probability of RVF being on the farm (occurrence). The other component was a multinomial logit that estimated the proportion (which was subsequently converted into the number) of animals on the farm that had no clinical signs of RVF, had an abortion, or died (severity). The outcome was the aggregate number of sheep, goats, and cattle, and we assumed that the three outcomes (no clinical signs, abortion, and death) were mutually exclusive within an individual ruminant. Hurdle models can be used to analyze data that have high rates of zeros [47]. We developed a mixture model that included a binomial hurdle with a multinomial model using a structure that allows us to evaluate hypothesized risk factors for RVFV presence as a separate process from severity, while retaining the capacity to estimate the full spectrum of severity. Hurdle models have been used in other disease systems to predict pathogen presence and severity. For instance, Yang et al. [48] used a hurdle logistic model to identify risk factors for the occurrence of bovine digital dermatitis on the farm and risk factors for the total number of cases on that farm; similarly, Jalava et al. [49] used a hurdle-truncated Poisson model to estimate occurrence and number of cases of verotoxigenic *Escherichia coli* in municipalities as a separate processes to identify both risk factors for occurrence in a region and risk factors for higher numbers of cases.

The risk factors in the model were selected to represent four processes that we hypothesized were associated with the occurrence and severity of RVF on a farm (Figure 5). We hypothesized that the severity of the clinical signs reported on the farm during 2010 would depend on two of the processes: (1) the amplification of the virus supported by large populations of vectors and (2) the susceptibility of the host population present. The risk factors selected to represent vector amplification were the proportion of days with rain and the cumulative precipitation between 15 January and 15 March, 2010 and the number of natural water sources within access of the animals. The risk factors representing host susceptibility were the proportions of the farm’s ruminant population that were sheep and goats, the total number of domestic ruminants on the farm, and whether the farmer vaccinated any animals against RVFV before the outbreak started. These seven variables, representing vector amplification and host susceptibility, were used in both linear predictors that estimated the probability of abortion and death during the 2010 RVF outbreaks. We hypothesized that the second two processes were associated with the occurrence of RVF on a farm, these processes were whether (3) the ecological factors linked to the RVF endemic cycle were present on the farm and whether (4) RVF was imported onto the farm. Thus, for the hurdle portion of the model, we selected risk factors linked to endemic RVF: farm size; the number of pans accessible to the ruminants; whether the ruminants mixed with wildlife and the cumulative rainfall between 15 October and 15 December 2009 (when the first floodwater *Aedes* spp. likely hatched); and factors linked to importation: the number of ruminants purchased and the distance from which they were purchased; and the production class of the farm (smallholder (reference), semicommercial, or commercial).

The hurdle portion of the model estimates h, the probability of RVF occurring on a farm (Equation (1)). It is defined through a logit transformation:(1)$$h=\frac{e^{\bar{H}}}{1+e^{\bar{H}}}$$
where $\bar{H}$ is estimated by the linear predictor with a Gaussian process (GP; Equation (2)):(2)$$\bar{H}=\eta_{0}+\eta_{1}L+\eta_{2}K+\eta_{3}C+\eta_{4}F+\eta_{5}Z+\eta_{6}M+\eta_{7}O+GP$$

The variables and the parameters for all equations are defined (Table 3). H is the predicted outcome, whether RVF occurred on the farm, and is a realization of a Bernoulli process (Equation (3)):(3)H~Bernoulli(h).

Uninformative priors were used for all of the parameters (Table 4).

A spatial, latent variable Gaussian process (GP), represented in the equation for $\bar{H}$ as GP, was used to account for the localized spatial patchiness of outbreaks that has been described as being associated with RVFV outbreaks [34] and seroprevalence studies [50]. The geographic positioning coordinates of each farm were converted into Universal Transverse Mercator (UTM) coordinates and scaled by the furthest distance between any two farms. We used a GP with an exponentiated quadratic kernel and uninformative priors for the length scale (ρ), marginal standard deviation ($\alpha_{GP}$), and latent GP value ($\sigma_{GP}$) (Table 4). The GP effect was incorporated as a vector to be added to the linear predictor for $\bar{H}$.

The multinomial portion of the model estimated three proportions that sum to one. These include the proportion of ruminants that did not have clinical signs of RVF (N), the proportion that died (D) (Equation (4)), and the proportion that aborted (A) (Equation (5)). Here, $\bar{N}$ was used as the reference outcome and $\bar{D}$ and $\bar{A}$ are estimated by linear predictors (see Table 3 for definitions), given by
(4)$$\bar{D}=\delta_{0}+\delta_{1}S+\delta_{2}G+\delta_{3}T+\delta_{4}V+\delta_{5}W+\delta_{6}R+\delta_{7}P$$
(5)$$\bar{A}=\alpha_{0}+\alpha_{1}S+\alpha_{2}G+\alpha_{3}T+\alpha_{4}V+\alpha_{5}W+\alpha_{6}R+\alpha_{7}P.$$

N,D, and A represent the elements of the vector θ, which is a realization of a multinomial process that uses a normalized exponential function (accessed by calling the softmax function in Stan, and referred to below as softmax) and takes the outputs of the linear predictors (Equations (4) and (5)) and generates three proportions that sum to one, where
(6)$$\theta \sim Multinomial\left(softmax\left(\bar{N},\bar{D},\bar{A}\right)\right).$$

The probability of RVFV occurring on the farm and deaths and/or abortions occurring (y) was defined by the probability mass function for the hurdle multinomial model:(7)$$p\left(y|h,\bar{N},\bar{D},\bar{A}\right)=\{\begin{array}{c} \left(1-h\right)+\left(h\cdot Multinomial\left(y|Softmax\left(\bar{N},\bar{D},\bar{A}\right)\right)\right)\;if\;y=0,and\\ h\cdot Multinomial\left(y|Softmax\left(\bar{N},\bar{D},\bar{A}\right)\right)\;if\;y=1. \end{array}$$

We predicted the number of animals (X) that died, aborted, or had no clinical signs of RVF on each infected farm. An example of the equation that calculates the number of animals that died ($X_{D}$) is
(8)$$X_{D}=rbinom\left(1,T,\theta_{D}\right)\cdot rBernoulli\left(1,h\right).$$

### 4.6. Programming and Analysis

The model was written in Stan [51] and implemented in R 3.6.0 [52] through Rstudio 1.2.1572 [53] using the package rstan 2.18.2 [54]. Analyses and diagnostics were completed using ggmcmc 1.2 [55], coda 0.19.2 [56], tidybayes 1.1.0 [57], and bayesplot 1.7.0 [58] packages. Plots were made using ggplot2 3.2.0 [59]. All continuous data were standardized with the mean set to zero. The model was run with four chains of 5000 iterations, with a burn-in of 2500, and no thinning. The residuals and log-likelihoods were evaluated. The code to reproduce the analysis and a deidentified dataset are available at [60].

The convergence of the overall model was assessed by visually inspecting the trace plots for all parameters, evaluating the potential scale reduction factor ($\hat{R}$) and the effective sample size. Posterior predictive checks were evaluated for each set of outcomes. The residuals were mapped and visually checked for spatial structure (Appendix A). The median, 66% and 95% credible intervals (CrIs) of all parameter estimates were plotted. Where the credible intervals contained zero the proportion of the posterior that was above or below zero was calculated. Unlike binomial models, the sign and effect size of the parameter estimate of multinomial models are not sufficient alone to assess the relationship as the sign/direction can change across the range of the variable values. The changing sign is dependent on the nonlinear relationship between the risk factor and the other two dependent variables [61,62]. To confirm that sign of the estimates for the continuous parameters did not change with the variable, we calculated the predicted probabilities. This was done by calculating the expected probability for each of the three multinomial outcomes while one continuous variable was input per the original data and all other variables held to their means or reference values.

The Gaussian process was evaluated by identifying the distance that explained 99% of the covariance, which was calculated by estimating the covariance decay rate using
(9)$$Covariance=e^{-\left(\frac{scaled.distance^{2}}{2\cdot \rho^{2}}\right)},$$
where scaled.distance was a vector containing the distance between each pair of UTM coordinates among all farms. We identified the distance at which the covariance dropped below 1%.

We tested models with slightly different structures, for example, random effect to account for spatial structure, different representations of the ruminant population structure on the farm, different time periods for the rainfall variables, and so forth (Appendix A). These models were assessed using the Watanabe-Akaike Information Criterion (WAIC; Appendix A). As none of the models were clearly better (all WAIC values were within the standard error of the other models), the model that best represented our current understanding of farm-level risk factors for RVF was selected based on expert opinion. Prediction accuracy was assessed using the area under the curve (AUC).

## 5. Conclusions

Here we estimated that the proportion of animals that farmers reported dying/aborting during the 2010 RVF regional outbreaks in South Africa was lower than is typically reported for RVF outbreaks in the literature. Using these data, we specified a Bayesian multinomial hurdle model to identify risk factors of the number of deaths and abortions that occurred during an RVF outbreak. Our results indicate that the percent of days with precipitation between 15 January and 15 March, 2010, the number of natural water sources, and the proportion of the herd that was goats were positively associated with increased numbers of deaths and abortions during the 2010 RVF outbreaks. In contrast, the total number of domestic ruminants was negatively associated with the number of deaths and abortions. Vaccinating before the outbreak was negatively associated with the number of deaths but was not with abortion, and the proportion of domestic ruminants that were sheep was positively associated with the number of abortions. We did not identify strong support for any potential risk factors of RVF occurrence on the farm, although, localized spatial correlation seems to have a strong effect on the occurrence of RVF. This suggests that while short-distance transmission of RVFV between farms is important, other farm-level risk factors that could help farmers assess their farm’s individual risk of an RVF outbreak need to be better elucidated.

## Figures and Tables

**Figure 1 pathogens-09-00914-f001:**
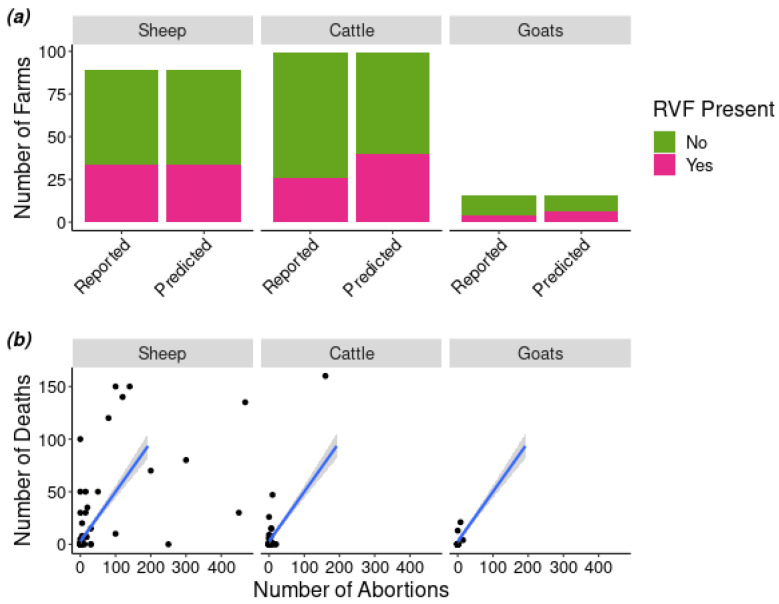
(**a**) The number of farms reported vs. the number predicted by the multinomial hurdle model to report an outbreak of Rift Valley fever (RVF) in a given species. (**b**) The number of species-specific deaths and abortions in 2010 as reported per farmers (black dots) and the relationship between the number of deaths and abortions predicted by the model for each species (blue line) with the confidence interval (grey ribbon).

**Figure 2 pathogens-09-00914-f002:**
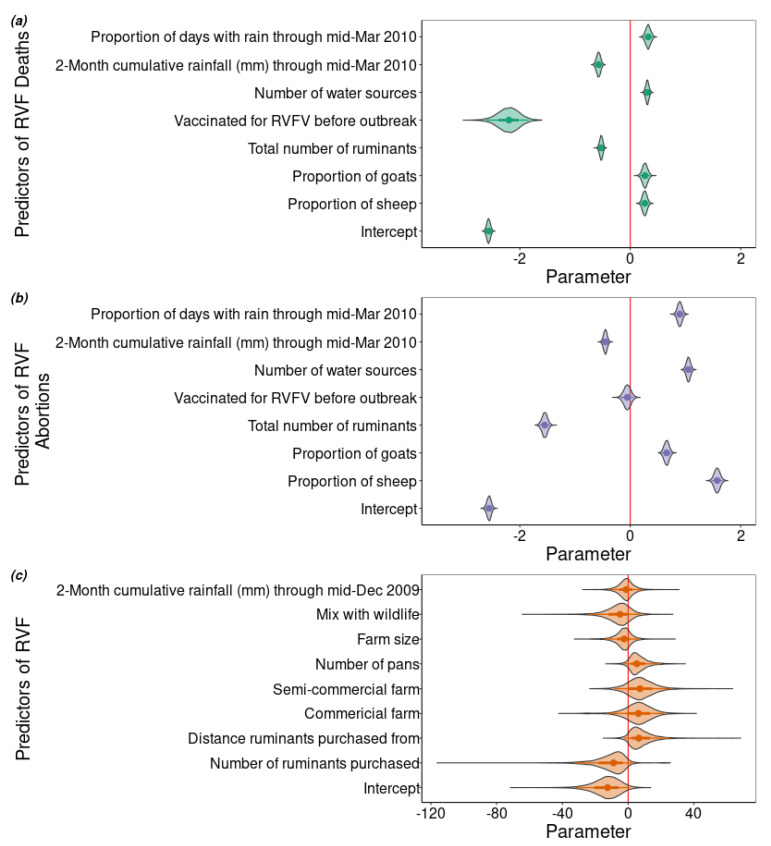
The standardized model parameters estimated as part of the (**a**) linear predictor for RVF-associated deaths, (**b**) the linear predictor of RVF-associated abortions, and (**c**) hurdle linear predictor for the occurrence of Rift Valley fever virus (RVFV). The points represent the median estimate of the posterior with the 66% (thicker) and 95% (thinner) credible intervals indicated by the line. A vertical line in red indicates an estimate of zero.

**Figure 3 pathogens-09-00914-f003:**
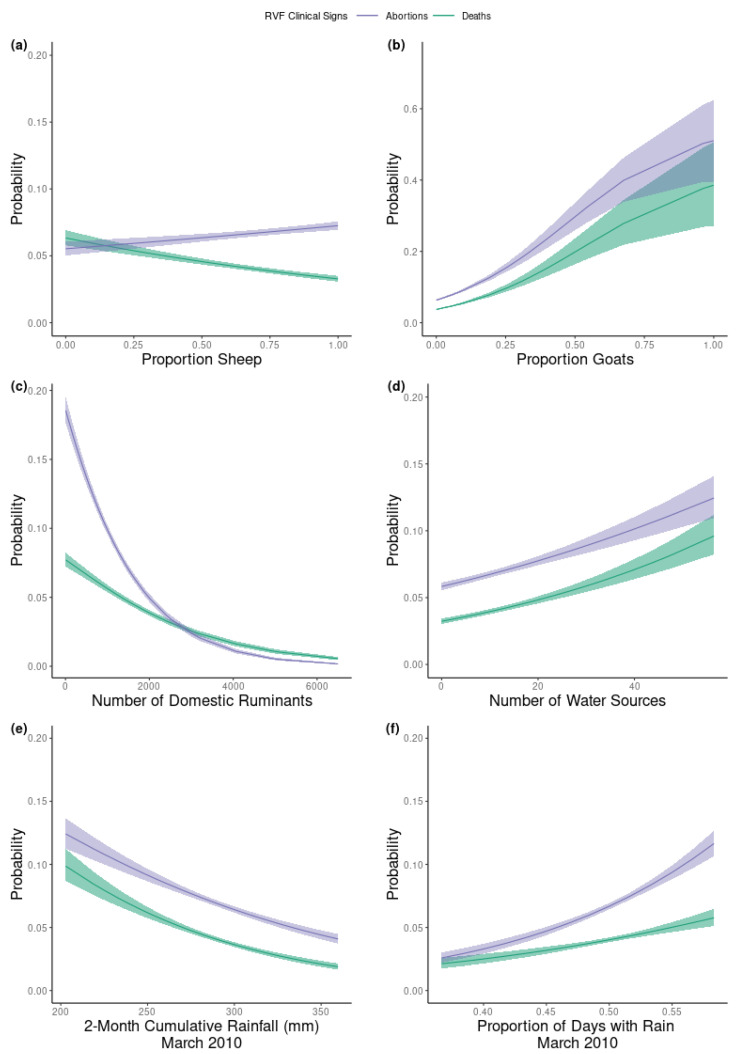
The predicted probabilities and 95% credible interval for death (turquoise) and abortion (purple) occurring on the farm during the 2010 RVF outbreaks for the six continuous risk factors in the model. The scaled parameters included (**a**) the proportion of the herd that are sheep; (**b**) the proportion of the herd that are goats; (**c**) the total number of ruminants on the farm; (**d**) the number of water sources available to the ruminants; (**e**) the two-month cumulative precipitation between 15 January and 15 March, 2010; (**f**) the proportion of days between 15 January and 15 March 2010 during which precipitation occurred.

**Figure 4 pathogens-09-00914-f004:**
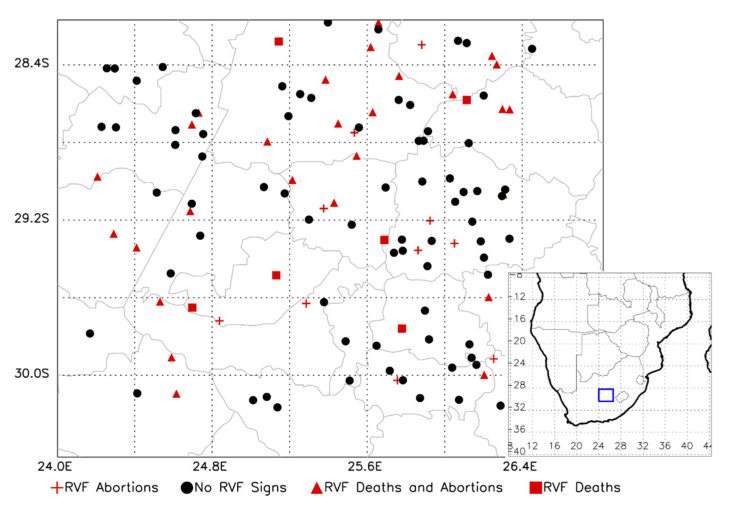
Map of the study area showing the participating farms and the RVF status reported.

**Figure 5 pathogens-09-00914-f005:**
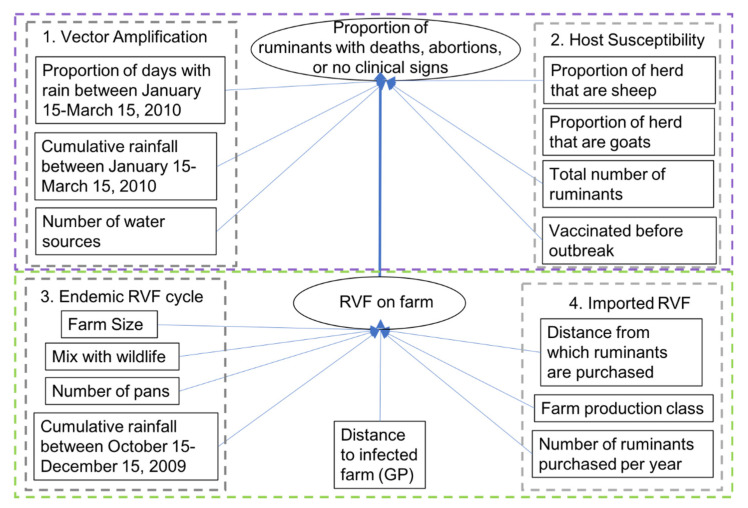
Diagram of model used in the analysis. The boxes represent variables in the model for which we have data. The circles represent the latent state variables. Above, the violet box contains the variables included in the two linear predictors that compose the multinomial logit part of the model and predicted the number of abortions, deaths, and animals with no clinical signs on each farm. We hypothesized that risk factors that supported (**1**) vector amplification on the farm or were related to (**2**) host susceptibility would be important in determining severity. The two linear predictors representing the multinomial part of the model use the same set of seven variables. The binomial hurdle portion of the model is below (green box) and contained the variables in the linear predictor that estimated whether RVF occurred on a farm. Two types of risk factors were hypothesized to contribute to whether RVFV occurred on the farm: (**3**) those that are linked to the ecology of the endemic RVF cycle and (**4**) those that contribute to the risk of importing RVFV (e.g., RVFV is brought onto the farm through the import of an infected animal). We also accounted for spatial variation based on distance between farms as RVF outbreaks occurring on nearby farms are associated with risk of an RVF outbreak.

**Table 1 pathogens-09-00914-t001:** The median, sample size, and range (percent, sample size, and 95% confidence interval) of the characteristics of the farms. For variables that represent counts, only the farms that reported having that species or farm feature were included in the estimate.

**Variable**	**Median**	**Number of Farms (*n* = 120)**	**Range**
Number of cattle in 2010	70	99	[2, 2500]
Number of sheep in 2010	300	89	[5, 6000]
Number of goats in 2010	50	16	[6, 400]
Total Number of ruminants in 2010	365	120	[5, 6500]
Proportion of sheep in 2010	0.923	89	[11, 100]
Proportion of goats in 2010	0.321	16	[0, 100]
Number of ruminants purchased	55	6	[1, 200]
Purchase distance	120	5	[12.5, 600]
Number of pans accessible to ruminants	1.5	51	[0.5, 7]
Number of water sources accessible to ruminants	6	110	[0.5, 56.5]
Farm size (km^2^)	1200	120	[0, 15,000]
Proportion of days with precipitation between January 15–March 15 2010	0.508	120	[36.7, 58.3]
2-Month cumulative precipitation between January 15–March 15 2010 (mm)	287.4	120	[202.8, 359.8]
2-Month cumulative precipitation between October 15–December 15 2009 (mm)	269.8	120	[217.1, 381.7]
	**Percent**	**Number of Farms Reporting**	**95% Confidence Interval**
Livestock mix with wildlife	49.2	120	[40.1, 58.2]
Vaccinated before RVF outbreak in 2010	25.8	120	[18.4, 34.3]
Vaccinated during RVF outbreak in 2010	50.8	120	[41.8, 59.9]
Affected by RVF outbreak in 2010	36.7	120	[28.2, 45.7]
Any reported deaths from in 2010	10.8	120	[6, 17.3]
Any reported abortions in 2010	8.3	120	[4.2, 14.3]
Small holder production	15	120	[8.6, 21.4]
Semi-commercial production	13.3	120	[7.3, 19.4]
Commercial production	71.7	120	[63.6, 79.7]

**Table 2 pathogens-09-00914-t002:** The median estimates and 95% credible intervals (95% CrI) for all of the parameters in the model, where the continuous parameters are unstandardized and in their original units. Estimates from the linear predictor for abortions are indicated by (A), those from the linear predictor for death are marked by (D) and those from the linear predictor of RVF on the farm, the hurdle portion of the model, are indicated by (R). GP = Gaussian process.

Variable	Median Estimate	95% CrI
Abortion Risk Factors		
Intercept (A)	−2.56	[−2.64, −2.48]
Percent of sheep (A)	1.24	[1.19, 1.28]
Percent of goats (A)	0.16	[0.14, 0.18]
Number of domestic ruminants (A)	−932.24	[−1041.87, −824.5]
RVFV vaccinated (A)	−0.05	[−0.18, 0.07]
Number water sources (A)	18.98	[18.21, 19.74]
Two-month cumulative rainfall through mid-March 2010 (A)	272.83	[270.61, 274.95]
Percent of day with rain through mid-March 2010 (A)	0.54	[0.54, 0.55]
Mortality Risk Factors		
Intercept (D)	−2.57	[−2.63, −2.5]
Percent of sheep (D)	0.69	[0.66, 0.73]
Percent of goats (D)	0.09	[0.08, 0.11]
Number of domestic ruminants (D)	199.78	[138.6, 259.08]
RVFV vaccinated (D)	−2.2	[−2.61, −1.84]
Number water sources (D)	11.38	[10.8, 11.96]
Two-month cumulative rainfall through mid-March 2010 (D)	268.62	[266.1, 271.06]
Percent of day with rain through mid-March 2010 (D)	0.52	[0.51, 0.52]
RVF on the Farm Risk Factors (Hurdle)		
Intercept (R)	−12.5	[−32.35, 1.02]
Ruminants purchased (R)	−235.42	[−879.12, 65.25]
Purchase distance (R)	390.07	[−141.07, 1495.47]
Commercial farm (R)	6.29	[−13.09, 22.12]
Semicommercial farm (R)	7.18	[−8.23, 25.68]
Accessible pans (R)	9.2	[−2.68, 31.43]
Farm size (R)	−5165.69	[−42,737.92, 24,805.48]
Wildlife (R)	−4.97	[−23.87, 7.72]
Two-month cumulative rainfall through mid-Dec 2009 (R)	123.27	[−54.55, 299.38]
GP length scale (R)	0.02	[0.01, 0.02]
GP variance (R)	1158	[134.5, 14,508.57]

**Table 3 pathogens-09-00914-t003:** The list of variables and parameters used in the model and their definitions. For variables defined by data, the hypothesized process it represents is given by number, where (1) vector amplification on the farm; (2) host susceptibility; (3) ecology of the endemic RVF cycle; and 4) RVFV importation.

Variable/ Parameter	Definition	Process
	Parameters Estimated by the Model:	
y	The probability of RVFV occurring on a farm and an outbreak (death(s) or abortion(s)) occurring	
h	The probability of RVF occurring on a farm	
H	The latent variable representing whether RVF occurred on the farm (0 or 1)	
$\bar{H}$	The estimate of the linear predictor for RVF occurrence used in the logit transformation to calculate h	
N	The estimated proportion of ruminants with no clinical signs of RVF	
$\bar{N}$	The estimate of the reference category for RVF severity (no clinical signs) used in the multinomial transformation to N	
D	The estimated proportion of ruminants that died from RVF	
$\bar{D}$	The estimate of the linear predictor for RVF severity (deaths) used in the multinomial transformation to D	
A	The estimated proportion of ruminants that aborted from RVF	
$\bar{A}$	The estimate of the linear predictor for RVF severity (abortions) used in the multinomial transformation to A	
θ	A vector that contains the three proportions N,D, and A, which all sum to one	
$\eta_{X}$	A vector of parameters for the binomial hurdle linear predictor of RVF on the farm	
$\delta_{X}$	A vector of parameters for the multinomial linear predictor for RVF deaths	
$\alpha_{X}$	A vector of parameters for the multinomial linear predictor for RVF abortions	
GP	The Gaussian process	
	Variables Representing Data	
P	The proportion of days with precipitation between 15 January and 15 March, 2010	
R	The cumulative precipitation between 15 January and 15 March, 2010	(1)
W	The number of water sources (excluding troughs) within access of the ruminants	(1)
S	The proportion of ruminants that were sheep in 2010	(2)
G	The proportion of ruminants that were goats in 2010	(2)
T	The total number of ruminants on the farm in 2010	(2)
V	Whether the farmer vaccinated any ruminants against RVFV before the RVF outbreak started	(2)
Z	The farm size (km^2^)	(3)
M	Whether the domestic ruminants mix with wildlife	(3)
F	The number of pans within access of the ruminants	(3)
O	The cumulative rainfall between 15 October and 15 December, 2009	(3)
L	The number of ruminants purchased in the past year—assumed to be constant across all years	(4)
K	The distance from which ruminants were purchased—assumed to be constant across all years	(4)
C	The production class of the farm (smallholder (reference), semicommercial, or commercial)	(4)

**Table 4 pathogens-09-00914-t004:** Parameters defining the model specifications and prior distributions.

Parameter	Distribution	Prior Values	Definition
$\eta_{X}$	~StudentT(υ,μ,σ)	υ=3 μ=0 σ=10	A vector of parameters for the hurdle binomial portion of the model.
$\delta_{X}$	~StudentT(υ,μ,σ)	υ=3 μ=0 σ=10	A vector of parameters for the linear predictor of death due to RVF
$\alpha_{X}$	~StudentT(υ,μ,σ)	υ=3 μ=0 σ=10	A vector of parameters for the linear predictor of abortion due to RVF
Parameters related to the spatial component			
ρ	~InverseGamma(α,β)	α = 1.839943 β= 0.114937	The length scale—the scale at which distances are measured among inputs [63]
$\alpha_{GP}$	~HalfStudentT(υ,μ,σ)	υ=3 μ=0 σ=10	The marginal standard deviation corresponding to how much of the variation is explained by the regression function [63]
$\sigma_{GP}$	~Normal(μ,σ)	μ=0 σ=1	The latent GP, which is used to generate a multivariate normal vector [63]
Parameters related to the random effect model structure that was only used in model 2 (Appendix A)			
ω	~HalfStudentT(υ,μ,σ)	υ=3 μ=0 σ=10	A normalized vector of the farm-specific random effect
$\sigma_{RE}$	~Normal(μ,σ)	μ=0 σ=1	Standard deviation of the random effect term

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
