# Peer review of "Farm-Level Risk Factors of Increased Abortion and Mortality in Domestic Ruminants during the 2010 Rift Valley Fever Outbreak in Central South Africa"

_pathogens, 2020, doi:10.3390/pathogens9110914_

Round 1
Reviewer 1 Report
The study of Rostal et al. aims at identifying factors associated with RVF occurrence and severity at the farm-level, during the 2010 epidemics in South Africa. The study focuses on the central region of the country, which has been the most affected during the epidemic. To collect data (incl. disease data), they interviewed 120 farmers, several years after the epidemic (in 2015 and 2017). The authors developed a statistical model to account for multiple outcomes, and conditional on reporting a RVF outbreak, together with a spatial component. The authors found that rainfall and water sources related parameters were associated with increased severity, as well as an increased proportion of goats. The paper is well written, although I believe that having the Results part before the Methods here is confusing.
The results are not ground-breaking and some of them are difficult to interpret. This may be explained by several reasons: a recall bias (several years between epidemic and survey), a small sample size, but also inherent to the type of model used and assumptions.
However, this study represents a lot of work (from field data collection to modelling), and studies at the farm-level are not numerous. The model developed is interesting and novel to the field of RVF studies (spatial and multiple outcomes).
I believe that the paper deserves publication if the authors could clarify some major concerns. See comments below.
One main point is about the outcomes definitions and the justification of the model:
Does ‘RVF outbreak on a farm yes/no’ relate to an emergence on the farm yes/no, defined by seeing animals with symptoms (abortions/deaths)? If so, it means that if there are no abortions nor deaths then the farm is assumed RVF negative. A farm without deaths or abortions could be infected by the virus, showing other mild symptoms. This may depend for example, on the number of hosts, and the species on the farm and the relative proportion. Similarly, the rainfall the month prior to the farm being positive could also be interpreted as ‘emergence’ variable (RVF yes/no), therefore related to RVF outbreak yes/no.
If my interpretation of these assumptions is correct, then I find it risky to give some hierarchy on how explanatory variables are used with regards to the different outcomes. Since the case definition is not that clear cut, I question what is the value of using such a hurdle model (with no identified risk factors for a farm being classified 1/0) instead of a zero-inflated or a poisson multinomial, in which all explanatory variables could be considered the same way for example?
Specific comments:
Abstract: ‘This suggest that farmers have limited tools to assess their individual risk’ . I do not agree on the link between the results and this statement, and this does not seem to be the major point of the discussion. May be rephrase or replace this sentence.
-See also comments on the Discussion paragraph -
Introduction
Lines 76-89 look like an abstract. Instead, the objectives, with a few lines on the methods should appear.
Results
Why not presenting first the results related to the presence of an outbreak on the farm yes/no, and then the factors associated with severity (number of deaths or abortions)? (swap 2.3 and 2.2?).
Check for any spatial structure in the residuals (+ incl. a map to include in SI if relevant?)
Discussion
Was the species mixing within a herd accounted for, or were species considered separately for three different species?
Line 243: The type of RVF vaccine used in 2010 in South Africa is not available? Do you mean it could be the smithburn, or the inactivated, and this would vary across farms ?
Lines 286-287 suggest that because the analysis did not find any factors associated with outbreak occurrence, then farmers are not aware of the risks. Is this correct? I am surprised by this since the variables explored with RVF outbreak occurrence were not too prone to bias (in Figure 5: farm size, mix with wildlife, number of pans, cumulative rainfall, distance purchase, farm production class, number of ruminants per yer).
Reporting a RVF outbreak (yes/no) was not associated to any risk factors, therefore, why is the hurdle necessary (versus zero-inflated)?
In the light of all the study limitations, a paragraph summarising/listing what type of data would be worth collecting during an outbreak to make this model readily available should another outbreak occur (in the SI, for example).
Methods
Figure 4: add a map of the country, to locate the study region on it.
Figure legend: it is difficult to read the colours of the different dots. May be try to use two colours: negative farms (black dots) and positive farms in red. Among the positive, use different symbols (as an example: a cross for deaths, a dot for abortions, and a star for farms with both).
Please justify the use of a hurdle model, and add references of other papers that has used it in a similar way - in paragraph 4.5 (esp. line 354)
Figure 5: show on the figure on what variable the spatial component is defined on
Line 402: add ‘(GP)’ after ‘Gaussian process’
Line 418 & Equation 7: ‘the expected value ‘ of what? where is ‘y’ defined?
Equation 8: what’s the subscript D of XD? If D is deaths and XD is the number of deaths, then in line 419-420, should be written . ‘An example of the number of animals that died’ ...
Table 3: add a column saying which variable is data or estimated from the data, and try to classify them by sections (outcomes related variables, and like in your Figure 5 – vector amplification, host susceptibility, endemic RVF cycle, Imported RVF).
Table 3: You can remove GP as it is only a notation from the text. GP is not a variable, correct?
Table 4: title could be ‘variables defining the model specification and prior distributions’. To clarify this table, you could split the table in ‘outcome variables’, ‘farm random effect’, and ‘variable related to the spatial component’
Line 446: ‘with slightly different structures (Table S1)’ : be more specific. What were you testing here?
Supp Information
Legend Figure S1: the colors of the lines in the legend is not very visible. It would be useful to add in the figure title the color names in brackets: ‘no clinical signs (black), death (green) and abortion (purple)’
Table S2: why all these different models were tested?
To illustrate that they are all roughly the same (and to support the ‘none were clearly better’ statement in lines 447-448), plot the WAIC value with their SE to show how they overlap (x-axis: model 1 to 6, and y-axis WAIC value and SE boundaries).
Reviewer 2 Report
The article is well written and the statistical analyses is very complete. The topic is interesting and the results are useful to take account in future research.
However there are a couple of issues that should be corrected:
1- The authors hypothesize that the amplification of the vectors (mosquito species) may play an important role in the transmission of the virus and they include this theory into the aims of the study. They repeat this parameter several times through the article (pag. 2 line 81, pag. 8 line 184, pag.8 line 213) however this parameter have been not included in the statistical analysis. In my opinion the among of water can not been used as marker of the vector amplification because there are not a prove correlation in this outbreaks, so this parameter should be eliminated.
2- The paragraph that spans from line 84 to line 89 describes the results, so I would remove it from the introduction.
In any case, the article is interesting and it is understandable that it is a retrospective study with the limitations of this type of work.
Round 2
Reviewer 1 Report
I have no further comments.